# Study to evaluate the effectiveness and cost-effectiveness of different screening strategies for identifying undiagnosed COPD among residents (≥40 years) in four cities in China: protocol for a multicentre cross-sectional study on behalf of the Breathe Well group

Zihan Pan [1,2] Andrew P Dickens [3] Chunhua Chi,[1] Xia Kong,[1] Alexandra Enocson,[3] Peymane Adab,[3] Kar Keung Cheng,[3,4] Alice J Sitch,[3] Sue Jowett,[3] Rachel Jordan,[3] on behalf of the Breathe Well Group

► Prepublication history and supplemental materials for this paper is available online. To view these files, please visit the journal online (http://dx.doi.org/10.1136/bmjopen-2019-035738).

For numbered affiliations see end of article.

**Correspondence to**
Professor Chunhua Chi;
chichunhua2012@qq.com and
Dr Andrew P Dickens;
A.P.Dickens@bham.ac.uk

## ABSTRACT

**Introduction** The latest chronic obstructive pulmonary disease (COPD) epidemiology survey in China estimated that there were 99 million potential COPD patients in the country, the majority of whom are undiagnosed. Screening for COPD in primary care settings is of vital importance for China, but it is not known which strategy would be the most suitable for adoption in primary care. Studies have been conducted to test the accuracy of questionnaires, expiratory peak flow meters and microspirometers to screen for COPD, but no study has directly evaluated and compared the effectiveness and cost-effectiveness of these methods in the Chinese setting.

**Methods and analysis** We present the protocol for a multicentre cross-sectional study, to be conducted in eight community hospitals from four cities among Chinese adults aged 40 years or older to investigate the effectiveness and cost-effectiveness of different case-finding methods for COPD, and determine the test performance of individual and combinations of screening tests and strategies in comparison with quality diagnostic spirometry. Index tests are screening questionnaires (COPD Diagnostic Questionnaire (CDQ), COPD Assessment in Primary Care To Identify Undiagnosed Respiratory Disease and Exacerbation Risk Questionnaire (CAPTURE), symptom-based questionnaire, COPD Screening Questionnaire (COPD-SQ)), microspirometer and peak flow. Each participant will complete all of these tests in one assessment. The primary analysis will compare the performance of a screening questionnaire with a handheld device. Secondary analyses will include the comparative performance of each index test, as well as a comparison of strategies where we use a screening questionnaire and a handheld device. Approximately 2000 participants will be recruited over 9 to 12 months.

**Ethics and dissemination** The study has been approved by Peking University Hospital and University of Birmingham. All study participants will provide written informed consent. Study results will be published in appropriate journal and presented at national and international conferences, as well as relevant social media and various community/stakeholder engagement activities.

**Trial registration number** ISRCTN13357135.

## Strengths and limitations of this study

► This is the first study to compare the effectiveness and cost-effectiveness of selected screening tests (questionnaires, peak flow meter and microspirometer) and strategies to screen for chronic obstructive pulmonary disease (COPD) in China.

► Recruiting participants from both urban and rural community hospitals will maximise the generalisability to primary care patients.

► Including four different screening questionnaires enables comparison of their test performance within a Chinese COPD population.

► Using blinded researchers to administer quality diagnostic spirometry minimises the risk of reviewer bias.

► The study will be conducted in four cities across China, which are geographically disparate but may not be representative of China as a whole.

## INTRODUCTION

Chronic obstructive pulmonary disease (COPD) is a common, preventable and treatable chronic condition characterised by persistent respiratory symptoms and airflow limitation.[1] Despite COPD being the third leading cause of death in the world,[2] COPD is underdiagnosed throughout the world due to multiple reason, including low awareness



of the disease and its consequences among the public and primary care health professionals, and the low use of spirometry.[3] While studies report the prevalence of undiagnosed COPD as being approximately 70% in Spain[3] and Poland[4] among those with the condition, a recent study in China reported that 96% of those with spirometry-confirmed COPD did not have a diagnosis.[5] Data from the US National Health and Nutrition Examination Survey revealed those with undiagnosed COPD were characterised by fewer symptoms,[6] this is reflected in China where 68% of undiagnosed people were asymptomatic.[7] What's more, about 30% of COPD patients were asymptomatic, those people were more likely to be underdiagnosed.[4 8]

The Global Initiative for Chronic Obstructive Lung Disease (GOLD) defines individuals as being at high risk of COPD if they have chronic respiratory symptoms, exposure to risk factors or medical/family history of respiratory disease.[1] According to the above definition, about 90% of people aged ≥40 years in China were at high risk of COPD in 2014.[9] The prevalence of diagnosed COPD in China was 13.7% in 2015.[5] Considering the substantial proportion of the Chinese population that is at risk of undiagnosed disease, screening for COPD in China is essential. Recently, China called for national policies and programmes for the prevention and early detection of COPD.[5 10] In line with this, government agencies have recommended the incorporation of pulmonary function tests into routine health examinations in China's thirteenth five-year plan for healthcare.[11]

While guidelines recommend that COPD is diagnosed based on spirometry and symptomology,[1] spirometry is not always available in primary care settings in China.[12 13] Among a large population of COPD patients in China, less than 12% had ever been tested using spirometry.[5 10] As a result, there is a need for simple and affordable COPD screening tools in primary care settings.

While COPD screening programmes are not currently recommended by the USA[14] or UK[15] due to insufficient evidence of health benefits, the national policy do recommend screening for undiagnosed COPD in China.[11] Despite the support for screening in China, there is no recommendation on the best strategy or approach to use. Multiple screening questionnaires have been developed to identify patients at risk of COPD, either in primary or secondary care settings.[16–20] Questionnaire items include the presence of respiratory symptoms (eg, wheeze, dyspnoea and cough) while some tools also explore exposures, smoking history and age. The questionnaires are all designed to be self-completed, but vary regarding the populations in which they were developed/validated, for example, general population or targeted groups such as symptomatic patients or current smokers. Microspirometers are small handheld devices that measure lung function, which are low cost, quick to use and require minimal coaching for patients. Peak flow monitors are simple, low-cost devices that measure how much air patients can expel during a forced expiration (peak expiratory flow, (PEF)),

and evidence indicates that these devices may also be suitable as a possible screening tool for COPD.[18 21 22]

Screening tests can be used individually or in combination as screening 'strategies'. Systematic reviews and more recent primary studies typically assess the use of a single test, concluding that many of the available tests may be appropriate for use in COPD screening.[23–25] However, studies in community settings in China are limited and it is not known which screening test or strategy would be most appropriate to use. As a middle-income country with a large potential COPD population, it is important to explore the most effective and cost-effective screening strategy. Accurately detecting individuals who merit referral for quality diagnostic spirometry could minimise the number of ineligible referrals, thus protecting health system resources and ensuring appropriate and timely treatment for those subsequently diagnosed.

## AIMS AND OBJECTIVES
### Aim
The aim of the study is to identify the most effective and cost-effective screening strategy for identifying undiagnosed COPD among those aged 40 years or older in China.

### Objectives
► To determine the comparative test performance of all screening tests and strategies in diagnosing COPD (confirmed by quality diagnostic spirometry).
► To evaluate the cost-effectiveness of each screening strategy.

## METHODS AND ANALYSIS
Study recruitment commenced in February 2019 and ended in December 2019.

### Design
Multicentre cross-sectional test accuracy study. The study is registered at http://www.isrctn.com.

The Standards for Reporting of Diagnostic Accuracy Studies guideline[26] was used for reporting studies of diagnostic test accuracy to inform the content of the protocol and we will use this to report the study.

### Study setting
The study will be implemented in four cities in China: Beijing (North), Chengdu (Southwest), Guangzhou (South) and Shenyang (Northeast). Cities were purposively selected to represent urban/rural settings and differing geographic areas of the country, where exposures, lifestyles and the prevalence of COPD may differ. The national study about COPD prevalence in 2007 in China was taken as a selection reference. Each selected city had the highest prevalence of COPD in each geographic area; prevalence is shown on the map.[27] Participants will be recruited from eight community health service centres (CHSC); one rural and one urban in each city. The study sites are shown on the map ([figure 1]).

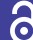

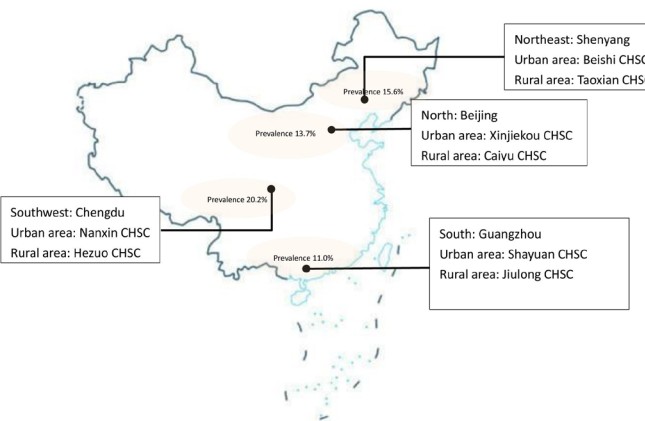

**Figure 1** The map of Breathe Well China research sites. CHSC, community health service centre.

## Study population
### Inclusion criteria
► Aged ≥40 years.
► Residing in the catchment areas of the participating CHSCs in the four cities.

### Exclusion criteria
► Unable to perform spirometry (eg, dementia or lack of teeth-cannot make a good seal).
► Contraindicated for spirometry (chest infection in the last 3 weeks, coughing up blood in the last month, severe angina, uncontrolled high blood pressure, pneumothorax or history in the last 3 months of tuberculosis, heart attack, detached retina or surgery on chest/abdomen/brain/ears/eyes).
► Currently pregnant/breastfeeding.
► Previous adverse reaction to salbutamol.

## Recruitment
Participants will be recruited to the study via two main routes, advertisement or doctor referral. Participating CHSCs and their satellite offices will advertise the study by displaying posters and sending messages to their secure/closed/other resident WeChat social media groups, inviting residents to contact the research team if they are interested in taking part. Potentially eligible patients visiting the participating CHSCs will be given a study information sheet by the healthcare professionals and invited to attend a study assessment with researchers. Study participants will also be encouraged to promote the study to their family members and friends. The recruitment route of all participants will be recorded.

For the first 4 weeks of recruitment, the study will only be conducted in Beijing to allow all study processes to be piloted and altered as required, after which it will be implemented in the other three cities. Recruitment flow through the study is summarised in figure 2.

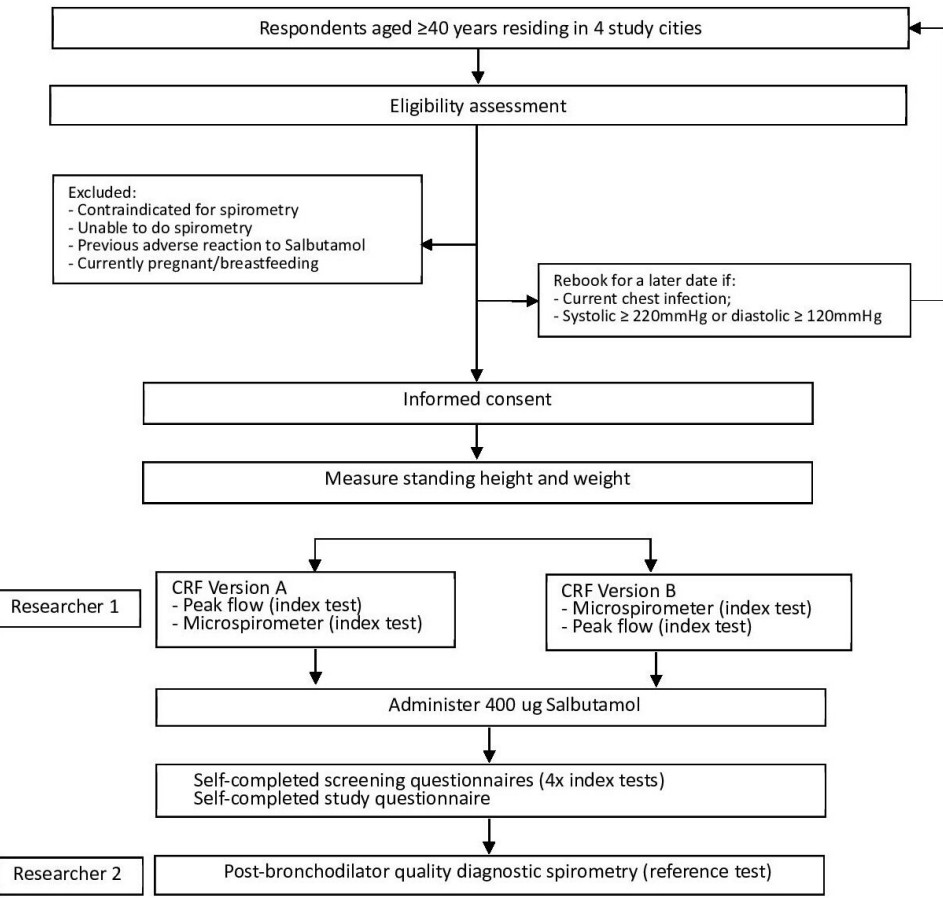

**Figure 2** Flow of participants. CRF, case report form.



## Study tests

The study will use a paired design, with all participants receiving the index tests and reference test during the same study assessment. The study will administer a total of six index tests (pre-bronchodilator peak flow and microspirometry and four screening questionnaires) and one reference test (post-bronchodilator quality diagnostic spirometry) to each participant.

### Index tests
#### Lung function test - peak flow

A trained researcher will assess PEF using a simple peak flow meter (USPE, China). Each participant will perform three blows without administration of bronchodilator, after which the researcher will record the highest PEF. For the main analysis, PEF rates of <350 L/min for men and <250 L/min for women will be used to indicate a positive test.[18]

#### Lung function test - microspirometer

Microspirometry will be performed with minimal coaching by a trained researcher using a simple handheld microspirometer (Vitalograph COPD6), to measure forced expiratory volume $(FEV)_1$, $FEV_6$ and $FEV_1/FEV_6$ ratio. Each participant will perform three blows using the device, after which the researcher will record the highest $FEV_1$ and $FEV_6$ values, and the $FEV_1/FEV_6$ ratio. For the main analysis, $FEV_1/FEV_6$ ratios of <0.75[28] and <0.78[29] will be assessed to indicate a positive test.

#### Screening questionnaires

Four screening questionnaires will be used in the study; the COPD Diagnostic Questionnaire (CDQ),[17 30] the COPD Screening Questionnaire (COPD-SQ),[19] a symptom-based questionnaire[31] and COPD Assessment in Primary Care To Identify Undiagnosed Respiratory Disease and Exacerbation Risk Questionnaire (CAPTURE)[18] (online supplemental appendix 1). The selection of questionnaires maximises symptoms being assessed and minimises duplication of items, while allowing comparison of the most relevant questionnaires. Recommended cut-points for each questionnaire will be used to identify those at risk of COPD for diagnostic spirometry, with potential additional analyses to explore optimal cut-points.

### Reference test

Post-bronchodilator quality diagnostic spirometry (20 to 60 min after administration of 400 µg salbutamol) will be performed by a trained researcher using a portable spirometer (Easy On-PC, NDD). Lung function data including $FEV_1$, forced vital capacity (FVC) and $FEV_1/$ FVC ratio will be recorded in the NDD software, and will also be imported to the study REDCap database. Accuracy of the device flow heads will be verified at the start of each assessment day by the researchers; calibration is not required. Participants will perform a maximum of six blows, or less if repeatability within 100 mL or 5% is achieved (Association for Respiratory Technology and Physiology standards, 2013).[32] For the purposes of

this study, a COPD diagnosis will be defined as airflow obstruction based on the lower limit of normal using the Global Lung Initiative equations, according to post-bronchodilator quality diagnostic spirometry.

### Ordering of assessments

Index tests will be conducted before the reference test for all participants, and the reference test will be administered by a different researcher who will be blind to the previous test results. To decrease the potential training effect within the index tests, the order of the peak flow and microspirometer will be alternated, that is, approximately half of the participants will perform peak flow first and vice versa. The screening questionnaires will always be completed after administration of salbutamol, during the 20 to 60 min time frame permitted prior to the reference test. Due to use of pre-printed study material, the order of the screening questionnaires will not be alternated.

Besides that, participants' standing height (stadiometer) and weight (scales) will be measured. Participants will also be asked to complete a study questionnaire by themselves, in a separate area of the assessment clinic, and return the completed forms to the researcher. The study questionnaire (online supplemental appendix 2) will include items relating to the following topics: demographic data (sex, age, marital status, education level and deprivation); smoking status; exposures (biomass smoke, occupational exposure to chemicals and particulates); health (medical diagnoses (including COPD, asthma, TB and so on), comorbidities and respiratory symptoms); quality of life (COPD assessment test). A member of the research team will be available to help participants to complete questionnaires if necessary.

### Data collection
#### Study assessment

The study assessment will last approximately 80 min, including six stations. There will be two researchers at each assessment clinic to enable the assessments to run in parallel, and all data will be recorded on case report forms (CRFs), ensuring standardised data collection/ recording.

At the end of the study assessment, researchers will provide all participants with information about the level of their airway obstruction, suggest they contact a doctor if appropriate and answer any immediate questions they may have. The flow of participants is presented in figure 2.

#### Resource use data

To calculate the healthcare costs of delivering each screening strategy, we will determine the unit costs and quantity of any equipment, medication and consumables required, as well as staff type and grade, staff time taken to deliver each individual test and use of facilities. The staff time taken will be collected with a simple questionnaire for researchers to fill in for each test (online supplemental appendix 3). Equipment costs (peak flow meters and spirometers) will be amortised over the estimated

lifespan of the equipment. The cost per patient visit will be calculated using assumptions regarding the total number of patients the equipment will be used for. In addition, each individual test will be timed at a sample of assessment clinics so that an overall mean time and range for each test can be estimated.

## Statistical methods
### Sample size
The Alonzo method for paired test accuracy studies[33] was used to calculate the sample size, assuming independence of tests and a prevalence of 12%, we will have 90% power to detect a difference in sensitivity of 10% (95% vs 85%[18 30 34 35] with 1622 participants). If the sensitivity of tests is slightly lower in this population (90% vs 80%), we would have 90% power to detect this difference with a larger sample of 2279 participants.

### Analysis plan
Data will be analysed using Stata V.15.

Our primary analysis will compare the performance of a screening questionnaire (CAPTURE) with a handheld device (peak flow meter). Secondary analyses will include the comparative performance of each index test, as well as a comparison of strategies where we use a screening questionnaire and a handheld device.

The performance of each index test when diagnosing COPD (confirmed by quality diagnostic spirometry) will be investigated by presenting 2×2 tables and calculating the sensitivity, specificity, positive predictive value and negative predictive value, along with 95% CIs. For the tests with a continuous score, receiver operator curve analysis with area under the curve (with 95% CIs) will be produced. Comparisons of test accuracy between different index tests and different test strategies will be conducted using McNemar's test and logistic regression modelling.

Sensitivity analyses will explore the impact on test performance of the index tests and strategies when using different definitions of COPD, including (i) a combination of spirometry data and clinical confirmation, and (ii) using the GOLD definition (fixed ratio ($FEV_1$/FVC <0.7)) of airflow obstruction. Additional sensitivity analyses may explore the impact of spirometry quality as well as exploring optimal cut-points for the screening tests, in recognition that test performance will be dependent on the cut-points used.

A fully incremental cost-effectiveness analysis will be undertaken from a healthcare perspective to calculate the cost per true case detected for all pre-determined strategies. The strategies (including combinations) will be ordered by the number of true cases detected, from least to greatest, and the principles of dominance and extended dominance will be applied to eliminate redundant strategies from the analysis. Sensitivity analysis will be undertaken to explore the impact on results of any changes in assumptions, for example, time taken for a strategy.

## Training
A 2-day training event will be organised for all researchers to ensure standardised study processes are followed at all research sites. Training will cover study processes and assessment techniques as well as expert teaching regarding respiratory physiology and spirometry lung function tests. Researchers' competency in conducting spirometry will be certified at the end of the training. Spirometry traces from practice sessions will be over-read by an expert to ensure sufficient quality prior to participant recruitment commences. Local respiratory specialists will over-read all spirometry tests during the study period, to ensure quality is maintained. During site initiation visits, the study team will observe a complete study assessment to ensure researchers adhere to the study protocol. The study will conduct monitoring site visits throughout the study period.

## Patient and public involvement
The research team conducted a research prioritisation exercise with patients, clinicians and policy makers, and the need to identify effective screening strategies for undiagnosed COPD was one of the research areas prioritised. All stakeholders involved in this exercise will receive study updates twice a year, will be kept informed of findings and will be consulted at the end of the study regarding implications for practice and policy decisions, as well as advice on appropriate dissemination of study findings.

A patient advisory group (PAG) has been set up, which is funded to meet at approximately quarterly intervals or according to need, and will advise on a range of aspects of the design, conduct, analysis and dissemination of the study. The PAG will discuss issues as requested by the CIs and the chair will report their comments back to the investigators.

In addition, the study has a trial steering committee (TSC) that meets regularly and comprises various independent members, including a patient and a clinician representative as well as international experts in respiratory research. The TSC also includes several members of the study research team.

## ETHICS AND DISSEMINATION
### Ethics and informed consent
The study has been approved by the Peking University First Hospital (2018-R-141, PUFH) (online supplemental appendix 4) and the University of Birmingham (ERN_18–1177, UoB) (online supplemental appendix 5). Residents responding to the study invitation will be given the study information sheet with enough time to read it and will have opportunity to ask the researcher any questions about the study. Interested respondents who are eligible for the study will be asked to sign a consent form (online supplemental appendix 6), or if unable to consent, a family member will be asked to sign on their behalf. Consent will also be sought to allow the research team to contact participants about future studies related



to the Breath Well programme; this is optional and will not affect eligibility for the study described in this paper.

### Indemnity
The study is not an intervention study, and as such poses low risk to participants. However, clinical insurance was purchased in case of serious adverse events.

### Data storage
Study data will be entered into a bespoke REDCap online database. All electronic data held by the research team will be password-protected and stored on encrypted study laptops. Paper-based data will be held in locked filing cabinets in the study office in each site. The research team will conduct monitoring visits of all research sites during the recruitment period to ensure data are being collected, entered and stored according to pre-specified study working instructions.

### Dissemination and publication policy
Study results will be published in peer-reviewed journals and presented at national and international conferences, as well as relevant community/stakeholder engagement activities. Participants who explicitly express a wish to be informed about the research outcome will be contacted and offered to receive an article or poster with a lay summary of the study.

### DISCUSSION
This study aims to identify the most effective and cost-effective screening strategies for identifying undiagnosed COPD in the primary care setting in China.

To the best of our knowledge, this is the first study to assess the accuracy of different COPD screening strategies, including screening questionnaires, peak flow and microspirometer measurement. This study is being conducted in a range of community hospitals from rural and urban areas which are broadly representative of primary care institutions in China. The planned cost-effectiveness analysis will calculate the cost per true case detected for each strategy, which will help inform decisions about the future feasibility of screening strategies within the primary care setting in China. This trial should inform primary care across China and elsewhere with similar healthcare systems, and help to direct current effort towards case-finding more efficiently.

While the study will be conducted in four purposively selected cities, it is possible that additional cities will be required to obtain a representative sample of the Chinese COPD population. However, increasing the number of study locations would have introduced difficulties such as training and monitoring study sites, thus we believe the selected cities represent an acceptable balance between study feasibility and representativeness. Furthermore, estimates of effectiveness are based on measurements undertaken under research conditions. While the screening tests are likely to be reproducible in routine practice, it is possible that peak flow and microspirometer measures could be done to a higher standard in research settings, leading to potential overestimation of effectiveness.

This study also helps building research capacity within primary care, as it is the first respiratory study for the participating community hospitals and the majority of general practitioner (GP) researchers being taught how to conduct high quality spirometry will have no prior experience and might have difficulty in understanding the research process.

Recent health policies have seen lung function testing being incorporated into a routine health examination programme among the general population, and objectives being set to increase the proportion of those over 40 years old received lung function tests from 7.1% in 2017 to 15% in 2020 and 25% in 2025.[36] Considering the increasing importance of lung function testing in China and the intensive spirometry training given to clinicians through this study, we believe this study could also help improve the quality of COPD management in primary care in China.

Considering that there is no 'GP first contact' in China yet, it is challenging to plan how best to attract people attending community hospitals and recruit them into the study. However, voluntary pulmonary function screening identifies high rates of undiagnosed asymptomatic COPD.[7] How to encourage residents to volunteer to participate in screening is also something we need to consider. It is also hard to anticipate residents' willingness to participate in this study and how participants will respond to the study measures. However, what is worth mentioning is that, besides posters, referral by doctors, friends or family members, and WeChat, a social media which has a prominence in Chinese society now, also plays an important role in the recruitment process to inform residents or disseminate the programme. Last but not least, it will be important to discuss how this approach can be rolled out from a trial setting into routine practice. Real world study may be the most appropriate method to make it clear how the validated screening strategy works in practices.

COPD screening is extremely important to China and its 99.9 million potential COPD patients.[5] This study will provide robust evidence about the effectiveness and cost-effectiveness of different COPD screening methods and strategies and confirm which the best COPD screening strategy is. The service might be a template for delivery of a procedural screening strategy that can reach large numbers of an under-recognized population. Although the long-term benefits of screening are still to be proven, this programme has capacity to contribute significantly to improving public health.

**Author affiliations**
[1]Department of General Practice, Peking University First Hospital, Beijing, China
[2]Department of Pulmonary and Critical Care Medicine, Peking University Third Hospital, Beijing, China
[3]Institute of Applied Health Research, University of Birmingham, Birmingham, UK

[4]General Practice Development and Research Centre, Peking University Health Science Centre, Beijing, China

**Acknowledgements** The views expressed in this publication are those of the author(s) and not necessarily those of the National Institute for Health Research (NIHR) or the Department of Health and Social Care. We gratefully acknowledge International Primary Care Respiratory Group for introducing us to the primary care networks involved in this study and for its continued facilitation of clinical engagement. This paper presents independent research supported by the NIHR Birmingham Biomedical Research Centre at the University Hospitals Birmingham National Health Service foundation trust and the University of Birmingham

**Collaborators** The Breathe Well group

**Contributors** ZP and APD wrote the protocol paper with input from all other authors. RJ led the design of the trial, with contributions and advice from all other investigators. CC, XK, PA and KKC contributed to decisions on outcome measures. CC and KKC advised on involving general practitioner practices, RJ, PA, AE and APD advised on lung function testing. APD and RJ designed the intervention. AJS and SJ designed the analysis plan and economic evaluation. CC was the local principal investigator. All authors have read and approved the final draft.

**Funding** This research was funded by the National Institute for Health Research (NIHR) NIHR global group on global chronic obstructive pulmonary disease in primary care, University of Birmingham, (project reference: 16/137/95) using UK aid from the UK Government to support global health research. The views expressed in this publication are those of the author(s) and not necessarily those of the NIHR or the UK Department of Health and Social Care.

**ORCID iDs**
Zihan Pan http://orcid.org/0000-0003-4502-1107
Andrew P Dickens http://orcid.org/0000-0002-7591-8129

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
