## [Reviewer comments · BMJ Open]

ARTICLE DETAILS

TITLE (PROVISIONAL)	A study to evaluate the effectiveness and cost-effectiveness of different screening strategies for identifying undiagnosed COPD amongst residents (≥ 40 years) in four cities in China: protocol for a multicenter cross-sectional study. On behalf of the Breathe Well group.
AUTHORS	Pan, Zihan; Dickens, Andrew; Chi, chunhua; Kong, Xia; Enocson, Alexandra; Adab, Peymane; Cheng, Kar Keung; Sitch, Alice; Jowett, Sue; Jordan, Rachel

VERSION 1 – REVIEW

REVIEWER	Job FM van Boven University Medical Center Groningen, Groningen Research Institute for Asthma and COPD (GRIAC), The Netherlands
REVIEW RETURNED	16-Jan-2020

GENERAL COMMENTS	In my opinion this is a well-designed study addressing an important issue, i.e. the underdiagnosis of COPD in China. I just have some suggestions to improve the reporting/writing. Abstract p.3.l.19. "In comparison with quality diagnostic spirometry". This is repeated two lines lower ("the reference test is quality diagnostic spirometry"). I think one can be deleted. Instead, adding some more information on the primary outcome and how the comparison is done would be helpful (as provided on page 8, line 50-54). p.3.l.28: relative=relevant? Introduction p.4.l.35: 30% of COPD p.4..42: denied=defined p.5.l.5 "but" can be deleted, recommendations=recommendation p.5.l.7: researches=studies, had=have p.5.l.23: recent=earlier (this study is from 2003, not really recent) p.5.l.31: settings=setting Methods p.5.l.41: Can the authors confirm that the study is not completed yet (this is a prerequisite for publication of BMJ Open protocols). p.5.l.46-56: Usually, the "Aims and objectives" are placed right after Introduction and are not part of Methods and analysis p.6.l.6: Most study protocols in BMJ Open should now be reported according to the SPIRIT checklist. Can this be added as attachment? p.6.l.19: city=area?
---

	p.6.l.28: Just as a comment (I understand this cannot be changed): should the patient not have any risk factor (smoking, air pollution exposure?) or symptoms? p.7.l.48: Maybe a sub-analysis can be considered using the fixed ratio (0.7)? p.7.l.60: Can the authors clarify whether the questionnaire is completed before or after the tests? Are these moments alternated as well? p.8.l.12: Here, a reference to the questionnaire (in appendix) can be considered. p.9.l.8: I assume the GOLD definition is equal to the fixed ratio (0.7), so would be good to specify. Discussion p.11.l.26: Given the willingness to participate is unclear, I recommend to measure the response rate. Important for generalizing the results and inform scale-up.
--	--

REVIEWER	Michael Crooks Hull York Medical School, UK
REVIEW RETURNED	27-Jan-2020

GENERAL COMMENTS	The described study is of international interest. I have the following comments:  - The introduction is difficult to read and in some parts does not make sense. For example, the use of undiagnosed and underdiagnosed appears inconsistent and a little confusing. Also, the text states that GOLD 'denied subjects' where I suspect it should read 'described subjects'. This section would benefit from revision. The other sections of the manuscript have occasional typographical errors. - It would help if the rationale for the 'index test' cut-off values could be described. It would also be useful to describe the potential impact of the cut-off values on the test performance. - This study is exploratory, analysing a range of questionnaires and lung function measures, either alone or in combination and comparing them with a gold standard. The cost-effectiveness analysis appears also to be largely theoretical, based on the assumed requirements for a screening programme using individual proposed index tests/combinations. I feel that it would be beneficial to highlight the anticipated strengths and weaknesses of the study design/analysis approach in this regard, with particular regard to generalizability.
---

VERSION 1 – AUTHOR RESPONSE

Reviewer(s)' Comments	
Reviewer: 1	Revision
Abstract p.3.l.19. "In comparison with quality diagnostic spirometry". This is repeated two lines lower ("the reference test is quality diagnostic spirometry"). I think one can be deleted. Instead, adding some more information on the	We have deleted "The reference test is quality diagnostic spirometry."

primary outcome and how the comparison is done would be helpful (as provided on page 8, line 50-54).	We have added the following text regarding the primary outcome and planned comparison: "The primary analysis will compare the performance of a screening questionnaire with a handheld device. Secondary analyses will include the comparative performance of each index test, as well as a comparison of strategies where we use a screening questionnaire and a handheld device."
p.3.l.28: relative=relevant?	Revision completed
Introduction p.4.l.35: 30% of COPD	Revision completed
p.4.42: denied=defined	Revision completed
p.5.l.5 "but" can be deleted, recommendations=recommendation	Revision completed
p.5.l.7: researches=studies, had=have	Revision completed
p.5.l.23: recent=earlier (this study is from 2003, not really recent)	Revision completed
p.5.l.31: settings=setting	Revision completed
Methods p.5.l.41: Can the authors confirm that the study is not completed yet (this is a prerequisite for publication of BMJ Open protocols).	Data collection was still ongoing when we submitted the paper, and it finished in December 2019. While we acknowledge our manuscript was submitted at the very end of the data collection period, we discussed the study timeframes with the Editor in advance and our submission was permitted.
p.5.l.46-56: Usually, the "Aims and objectives" are placed right after Introduction and are not part of Methods and analysis	Revision completed
p.6.l.6: Most study protocols in BMJ Open should now be reported according to the SPIRIT checklist. Can this be added as attachment?	As this is a diagnostic test accuracy study, the results are reported according to STARD. However, the protocol manuscript is formatted according to the SPIRIT checklist.
p.6.l.19: city=area?	Yes, different cities represent different areas. In this context, "area" refers to the four geographic regions of China (south China, north China, southwest China and northeast China).
p.6.l.28: Just as a comment (I understand this cannot be changed): should the patient not have any risk factor (smoking, air pollution exposure?) or symptoms?	It is important to explore the performance of screening tests within a primary care population, the majority of whom will have at least one risk factor for COPD. Within the results paper, we will describe the study

	sample characteristics so the reader can interpret the test performances in light of the participants we recruited. In addition, the screening questionnaires capture risk factors and symptoms, so this is part of the alternative strategies.
p.7.l.48: Maybe a sub-analysis can be considered using the fixed ratio (0.7)?	We agree that this will be important, and we did report a planned sensitivity analysis to explore this in the submitted manuscript. However, we have inserted additional test to clarify this within the Analysis plan section..
p.7.l.60: Can the authors clarify whether the questionnaire is completed before or after the tests? Are these moments alternated as well?	We have inserted text to clarify this within the 'Ordering of assessments' section.
p.8.l.12: Here, a reference to the questionnaire (in appendix) can be considered.	We have added a reference to the appendix, as suggested
p.9.l.8: I assume the GOLD definition is equal to the fixed ratio (0.7), so would be good to specify.	We have added text to clarify that the GOLD definition is equal to the fixed ratio of <0.7.
Discussion p.11.l.26: Given the willingness to participate is unclear, I recommend to measure the response rate. Important for generalizing the results and inform scale-up.	We agree with this suggestion. The response rate will be reported in the STARD recruitment flow chart in the main paper.
Reviewer: 2	
- The introduction is difficult to read and in some parts does not make sense. For example, the use of undiagnosed and underdiagnosed appears inconsistent and a little confusing. Also, the text states that GOLD 'denied subjects' where I suspect it should read 'described subjects'. This section would benefit from revision. The other sections of the manuscript have occasional typographical errors.	We have taken the opportunity to revise the Introduction, improving the overall flow and correcting terminology where required.
- It would help if the rationale for the 'index test' cut-off values could be described. It would also be useful to describe the potential impact of the cut-off values on the test performance.	For the screening questionnaires, we used the cut-points recommended in the development and validation papers. Regarding the peak flow meter and microspirometer, we adopted cut-points used in well-designed prior studies. References to the relevant papers were provided in the submitting manuscript so we hope it is sufficiently clear, but we would happy

	to take advice on how the wording can be improved. We have inserted additional text within the Analysis plan section, acknowledging the impact that selected cut-points will have on test performance.
- This study is exploratory, analysing a range of questionnaires and lung function measures, either alone or in combination and comparing them with a gold standard. The cost-effectiveness analysis appears also to be largely theoretical, based on the assumed requirements for a screening programme using individual proposed index tests/combinations. I feel that it would be beneficial to highlight the anticipated strengths and weaknesses of the study design/analysis approach in this regard, with particular regard to generalizability.	We will be estimating costs based on data collected during the study, and as such, we hope they reflect realistic costs for using these screening tests in primary care. Although population screening is theoretical in many countries, other settings, such as China, do have a policy to screen for undetected COPD. However, we accept that the study data may not be entirely generalizable to clinical practice, as the data will be collected in a research setting from four cities in China. We have inserted some text into the Discussion alluding to this potential limitation.

VERSION 2 – REVIEW

REVIEWER	Job van Boven University Medical Center Groningen, University of Groningen, The Netherlands
REVIEW RETURNED	23-Mar-2020
GENERAL COMMENTS	All my comments are addressed and I look forward to seeing the results of this study.
REVIEWER	Michael Crooks Hull York Medical School, UK
REVIEW RETURNED	17-Apr-2020
GENERAL COMMENTS	I feel that my comments from my earlier review have been adequately addressed. I look forward to reading the findings of this study.